# Lytic Polysaccharide Monooxygenase from *Talaromyces amestolkiae* with an Enigmatic Linker-like Region: The Role of This Enzyme on Cellulose Saccharification

**DOI:** 10.3390/ijms222413611

**Published:** 2021-12-19

**Authors:** Juan Antonio Méndez-Líter, Iván Ayuso-Fernández, Florian Csarman, Laura Isabel de Eugenio, Noa Míguez, Francisco J. Plou, Alicia Prieto, Roland Ludwig, María Jesús Martínez

**Affiliations:** 1Department of Microbial and Plant Biotechnology, Centro de Investigaciones Biológicas Margarita Salas, Spanish National Research Council (CSIC), Ramiro de Maeztu 9, 28040 Madrid, Spain; jmendez@cib.csic.es (J.A.M.-L.); lidem@cib.csic.es (L.I.d.E.); aliprieto@cib.csic.es (A.P.); 2Faculty of Chemistry, Biotechnology and Food Science, Norwegian University of Life Sciences (NMBU), 1462 Ås, Norway; ivan.ayuso-fernandez@nmbu.no; 3Department of Food Science and Technology, BOKU–University of Natural Resources and Life Sciences, Muthgasse 11, 1190 Vienna, Austria; florian.csarman@boku.ac.at (F.C.); roland.ludwig@boku.ac.at (R.L.); 4Instituto de Catálisis y Petroleoquímica, Spanish National Research Council (CSIC), Marie Curie 2, 28049 Madrid, Spain; noa.miguez@cisc.es (N.M.); fplou@icp.csic.es (F.J.P.)

**Keywords:** LPMO, wheat straw, brewers spent grain, saccharification, AA9, oxidative biomass degradation

## Abstract

The first lytic polysaccharide monooxygenase (LPMO) detected in the genome of the widespread ascomycete *Talaromyces amestolkiae* (TamAA9A) has been successfully expressed in *Pichia pastoris* and characterized. Molecular modeling of TamAA9A showed a structure similar to those from other AA9 LPMOs. Although fungal LPMOs belonging to the genera *Penicillium* or *Talaromyces* have not been analyzed in terms of regioselectivity, phylogenetic analyses suggested C1/C4 oxidation which was confirmed by HPAEC. To ascertain the function of a C-terminal linker-like region present in the wild-type sequence of the LPMO, two variants of the wild-type enzyme, one without this sequence and one with an additional C-terminal carbohydrate binding domain (CBM), were designed. The three enzymes (native, without linker and chimeric variant with a CBM) were purified in two chromatographic steps and were thermostable and active in the presence of H_2_O_2_. The transition midpoint temperature of the wild-type LPMO (Tm = 67.7 °C) and its variant with only the catalytic domain (Tm = 67.6 °C) showed the highest thermostability, whereas the presence of a CBM reduced it (Tm = 57.8 °C) and indicates an adverse effect on the enzyme structure. Besides, the potential of the different *T. amestolkiae* LPMO variants for their application in the saccharification of cellulosic and lignocellulosic materials was corroborated.

## 1. Introduction

One of the most important applications for plant biomass valorization is related to the production of second-generation ethanol, based on the degradation of lignocellulosic biomass, where the sources of fermentable sugars are cellulose and hemicellulose [1]. After deconstruction of the plant cell wall and removal of lignin [2], plant polysaccharides must be hydrolyzed into their corresponding monosaccharides by the concerted action of different types of enzymes, mostly glycosyl hydrolases (GHs). These enzymes catalyze the hydrolysis of the glycosidic bond between two monosaccharides, or between a carbohydrate and an aglycone [3].

Cellulose exploitation is particularly interesting, since it is the most abundant polysaccharide on the planet and the glucose derived from it can be used in a large number of applications. In general, it can be stated that all the enzymes that degrade cellulose exhibit a synergistic behavior, due to the fact that they work in a coordinated way [4,5]. However, cellulose has crystalline areas that are difficult for glycosyl hydrolases to access, and the model for the whole degradation of this polysaccharide seemed incomplete. The pioneer studies developed by Reese on cellulose degradation [6] suggested the possibility of the coexistence of a two-component system: (1) an unidentified component which would act in the first place reducing the crystallinity of the cellulose, and (2) the glycosyl hydrolases, that would degrade the polymer into glucose [7]. Several decades later, the first step was found to be due to the action of fungal and bacterial copper-containing lytic polysaccharide monooxygenases (LPMOs) [8]. These novel catalysts were included in the CAZy database [9], in the class of auxiliary activities, and currently belong to families AA9, 10, 11, 13, 14, 15 16, and 17. Enzymes from the AA9 family are N-terminal methylated enzymes, usually glycosylated, that act on β-1,4 linked glucoses and are found exclusively in fungi. The number of potential LPMO genes discovered in fungal genomes has increased exponentially in recent years, demonstrating that they are ancient enzymes and offer a vast new field to study cellulose degradation. However, despite the broad diversity of LPMOs, the number of enzymes isolated and characterized biochemically and catalytically up to date is still quite low.

LPMOs are monocopper enzymes in which copper is coordinated by a conserved histidine brace [10]. Their catalytic mechanism starts with the reduction of the solvent-exposed Cu^2+^ to generate reactive Cu^1+^ species by an external source of electrons. Cu^1+^ further reacts with the substrate using an *O*-containing co-substrate. It was generally assumed that LPMOs used O_2_ in a monooxygenase reaction, but the true nature of the co-substrate has been questioned recently since H_2_O_2_ outperforms O_2_ [11,12]. Either way, cellulose-active LPMOs will hydroxylate the C1 or C4 carbons that participate in the glycosidic bond between two glucose residues in the polymer. Cellulose-acting LPMOs extract a hydrogen atom from C1 or C4 at the cleavable glycosidic bond, followed by hydroxylation of the resulting radical substrate. Besides, some LPMOs oxidize both C1 and C4, which means they have a mixed product profile that includes double oxidation products [13]. This is an important feature since LPMOs are usually classified based on their regioselectivity. C1 oxidation generates an oxidized glucose which is a lactone with a nonreducing end. It is important to note, that, depending on the pH, this lactone is in equilibrium with the aldonic acid form, which predominates at a neutral pH. On the other hand, C4 oxidation leads to the formation of a reducing end, a 4-ketosaccharide which, in aqueous conditions, is converted into a gem diol. Double oxidized products in C1-C4 have both degradation products simultaneously (Figure 1).

As mentioned before, LPMOs need an external source of electrons to reduce the copper at the active site. Although in vitro experiments generally use ascorbic acid as the reductant molecule [7], in nature, a variety of phenolic compounds produced from lignocellulose degradation could serve as electron donors to LPMOs [14]. Besides, it was also postulated that LPMOs could act in coordination with cellobiose dehydrogenase as a natural electron donor [15]. Moreover, the synergistic action with other enzymes, such as formaldehyde oxidoreductase or the glucose–methanol–choline oxidoreductases [16], was also identified as a potential source of electron donors for LPMOs. Regarding the binding to cellulose, approximately 30% of LPMOs contain carbohydrate binding modules (CBMs) [17], which belong mainly to the CBM1 family in the case of fungal LPMOs [18]. In general, it is considered that CBMs play a crucial role to tether the catalytic domain containing the active site in close proximity with its substrate. This role could be even more important than in GHs, since LPMOs avoid oxidative self-inactivation. The biotechnological applications of LPMOs are a field in full expansion. In general, it is well proven that these biocatalysts are an important piece for cellulose deconstruction [19]: the addition of LPMOs to cellulolytic cocktails has improved cellulose hydrolysis, which has changed the vision on how to degrade this polysaccharide [8]. Until today, one of the main bottlenecks when working with LPMOs has been its functional characterization and the measurement of its physicochemical properties [20]. To overcome the limitations and difficulties for the detection of oxidized oligosaccharides [20,21], a few colorimetric assays have been developed [22,23] to easily follow LPMO activity. That is the case for the oxidation of 2,6-dimethoxyphenol (2,6-DMP) to the colored product coerulignone, using H_2_O_2_ as co-substrate [22].

In the above overall context, the ascomycete *T. amestolkiae* is being studied for its ability to secrete powerful cellulolytic enzyme cocktails when growing on different carbon sources [24], and many of the enzymes involved have been purified and characterized [25]. The analysis of its genome revealed that the number of genes encoding for putative enzymes implicated in plant cell wall degradation was much higher than for *T. reesei* and other cellulolytic model fungi [24]. It should be noted that the discovery and characterization of LPMOs from *Penicillium* or *Talaromyces* (its perfect form) species is currently very scarce, which is intriguing considering their excellent performance in the degradation of lignocellulosic biomass [26,27]. In this work, a gene that encodes for a putative LPMO in the genome of *T. amestolkiae*, as well as two variants of this enzyme, have been successfully expressed in *P. pastoris* and the recombinant proteins purified and characterized. In addition, another two variants of the LPMO were designed to compare properties. The role of these LPMOs for enhanced cellulose hydrolysis, as well as for the saccharification of lignocellulosic residues, have been studied.

## 2. Results and Discussion

### 2.1. Cloning, Production, and Purification of TamAA9A

In recent years, several studies have demonstrated that, in the presence of inducers of lignocellulolytic activities, the ascomycete *T. amestolkiae* produces a set of highly efficient glycosyl hydrolases (GHs), especially β-glucosidases and β-xylosidases [24,28,29,30,31,32]. These enzymatic cocktails rich in GHs were successfully applied to saccharify wheat straw [24] and the outstanding activity of the β-glucosidases secreted by this organism has been corroborated [28,29]. However, the potential of other interesting enzymes of this organism related to lignocellulose degradation, i.e., LPMO, remains unknown.

In a previous work, a single gene coding for a putative LPMO (TamAA9A protein) was detected in the annotated genome of *T. amestolkiae* (GenBank accession no. MIKG00000000) [24]. To confirm its expression, RNA from 7-day-old cultures of *T. amestolkiae* growing in Mandels medium, induced with crystalline cellulose as a carbon source, was extracted and subsequently retrotranscribed to obtain cDNA. The gene consisted of a sequence of 1060 bp, including one intron, and encoded a 333 amino acid protein (Appendix A). The peptide sequence of TamAA9A shows a high similarity to those of other AA9 fungal LPMOs, except for the C-terminal region. When modeled (Appendix A), this part of the protein was modeled as a random coil. This disordered structure is frequently found in cellulases, acting as a linker between the catalytic domain and a CBM [33]. The function of this “linker-like” domain in the LPMO of *T. amestolkiae* is quite uncertain, especially considering that the gene lacks a CBM sequence after the LPMO coding region. A very similar case was reported by Kojima et al. [34], when working with an AA9 LPMO from *Gloeophyllum trabeum*, in which alternative splice processing generated two variants of the enzyme, one of them with a C-terminal domain comprising approximately 55 residues without a predicted function. Although the exact role of this domain remains uncertain, a linker region that connects the catalytic domain and a CBM1 has been described in an LPMO produced by *Hypocrea jecorina* (HjLPMO9A) [35]. X-ray analysis of a truncated form of this enzyme demonstrated that the linker is an integral part of the structure of the catalytic domain of this LPMO, covering a hydrophobic patch [35].

To identify the role of the “linker-like” region in this LPMO, three different enzyme variants were constructed and expressed in *P. pastoris*: (i) the complete wild-type enzyme (TamAA9A-WT), including the catalytic domain and the “linker-like” region; (ii) only the catalytic domain (TamAA9A-CD) without the “linker-like” region; and (iii) a chimeric variant of the whole LPMO that incorporates the functional CBM of family 1, from a β-glucosidase of the same fungus [28] attached to the linker (TamAA9A-CBM).

Initially, the sequence of TamAA9-WT fused to the α-factor signal sequence, which usually enhances protein secretion, was expressed in *P. pastoris*. The transformants were isolated and grown in YEPS liquid medium for 5 days, monitoring the protein amount. However, no recombinant protein was detected by SDS-PAGE, which could be due to the ambiguous processing of the N-terminal sequence in *P. pastoris* when expressing a very high amount of enzyme [36]. At high expression levels, the proteins produced in this host are prone to have some additional amino acids in this region, due to protease miscleavage. Since N-terminal integrity has been proved to be very relevant for LPMO expression [20,37], the introduction of additional amino acids in this zone could produce inactive proteins. Then, the three LPMO variants with their native signal peptide of TamAA9A were cloned into the pPICZ vector, following the successful strategy performed by Kittl et al. for *Neurospora crassa* LPMO expression [23]. Samples taken from these cultures did not allow us to detect enzyme activity on 2,6-DMP oxidation, probably because enzyme activity could be inhibited by some medium components of the complex YEPS medium, as postulated for other fungal LPMOs [38]. However, the bands corresponding to the three recombinant versions of the LPMO were observed by SDS-PAGE analysis, confirming their secretion into the culture medium (Appendix A).

After this initial screening, the clones yielding the highest protein concentration were chosen for enzyme production in a 5-L bioreactor, monitoring the cultures for wet biomass and LPMO activity with 2,6-DMP as the substrate, and analyzing the supernatants by SDS-PAGE at the final incubation time (Figure 2A). After 4 days of fermentation a wet biomass of 209, 240, and 280 g/L for TamAA9A-WT, TamAA9A-CD and TamAA9A-CBM, respectively was determined. The bioreactor cultivations were carried out using a minimal medium, and the lack of interfering components allowed the successful monitoring of enzyme production by following the activity of the supernatants against 2,6-DMP (138, 171 and 240 U/L for TamAA9A-WT, TamAA9A-CD and TamAA9A-CBM, respectively). The three LPMO variants were purified to homogeneity after two chromatographic steps (hydrophobic interaction and anion exchange chromatography, Figure 2B,C), as corroborated by SDS-PAGE (Figure 2D).

### 2.2. Comparative Physicochemical Properties of TamAA9A Variants. Circular Dichroism and Thermostability

The molecular masses of the three variants, determined by SDS-PAGE, showed that all of them are glycosylated proteins, as reported for other recombinant proteins expressed in *P. pastoris* [39]. TamAA9A-CD showed a molecular mass close to 30 kDa (its theoretical mass was 24 kDa), which is consistent with 20% glycosylation. On the other hand, TamAA9A-WT and TamAA9A-CBM were massively glycosylated. The theoretical molecular masses for these variants (32 kDa and 34 kDa, respectively) contrasted with the empirical determinations (about 75–100 kDa), which implies around 150%–200% glycosylation (Figure 2D). These results suggest that additional glycosylation occurs in the “linker-like” region (which is absent in the TamAA9A-CD variant). Although such a high glycosylation degree is not common in other recombinant fungal LPMOs, the LPMO9H from *Heterobasidion irregulare*, also expressed in *P. pastoris*, was shown to have similar characteristics [40]. Analysis with NetOGlyc 4.0 indicated that the “linker-like” region of TamAA9A harbors 28 potential glycosylation sites, while the catalytic domain has only four. Enzymatic deglycosylation (of *N*-gylcosylation sites) using Endo H of the three variants (Appendix A) only showed a small reduction in protein mass, suggesting that most residues should be *O*-glycosylated, since enzymatic *O*-deglycosylation is not very efficient compared to *N*-deglycosylation [41].

Temperature and pH optima are fundamental parameters for an enzyme to be considered for industrial hydrolysis of lignocellulosic biomass. Generally, the biochemical characterization of LPMOs has been difficult due to the lack of easy methods to detect its activity [21]. The fact that the TamAA9A variants are able to oxidize 2,6-DMP made their characterization more feasible. First, the 24-h stability at pH values of 2–10 were analyzed at 4 °C, showing that all the variants had similar profiles, maintaining around 80% activity in the pH range of 4–7, and losing activity out of this range (Appendix A).

The three LPMO variants exhibited good thermostability for saccharification, which is usually conducted at 50 °C. However, while the T_50_ values of TamAA9A-WT and TamAA9A-CD were 76.1 °C and 74.4 °C, respectively, the T_50_ of TamAA9A was around 20 °C lower (57.9 °C). The reduced thermal tolerance of TamAA9A-CBM suggests that the CBM has a negative effect on its thermal stability. Considering that 2,6-DMP can be easily oxidized by different agents, including the copper released after LPMO inactivation, we decided to confirm the thermostability of the different proteins by circular dichroism, as previously reported for other LPMOs [42]. The change in the stability of the secondary structure started at about 60 °C, and it can be considered that all the variants lost their secondary structure at 75 °C (Appendix A). After adjusting to a sigmoidal equation, the transition midpoint temperature (T*m*) was 67.7 °C for TamAA9A-WT, 67.6 °C for TamAA9A-CD, and 57.8 °C for TamAA9A-CBM. These results confirmed the lower thermotolerance of the chimeric variant with the CBM and show that, even though the enzymes can still oxidize 2,6-DMP at a higher temperature, the unfolding of secondary structures begins a little earlier than calculated by determining the T_50_, especially for TamAA9A-WT and TamAA9A-CD.

Thermotolerance has been explored in other AA9 LPMOs. That is the case for LPMO9A from *Thermoascus auriantiacus* (TaLPMO9A), which retains 72% activity after a 45 min incubation at 100 °C [43]. The catalytic domains of TamAA9A and TaLPMO9A share a high sequence similarity, which could be related to their good thermal stabilities. Moreover, the thermostability of other LPMOs of the AA9 family, like those from *Scytalidium thermophilum*, *Malbranchea cinnamomea*, *Myceliophthora thermophila* [44,45], or *Talaromyces cellulolyticus* [46] are similar to that determined for TamAA9A. However, the most thermostable LPMOs are the bacterial LPMOAA10s found in a compost metagenome (mgLPMO10), with a melting temperature of 83 °C [47]. Despite this, due to the combination of thermostability and good pH tolerance of *T. amestolkiae* LPMO variants, we postulate they may be promising catalysts for lignocellulosic biomass valorization, since they are stable at the operational temperature used in industrial saccharification, where substrates are usually pretreated with alkali or diluted acid [48].

### 2.3. Substrate Specificity and Binding to Cellulose

As mentioned above, the discovery of the peroxygenase-like capacity of LPMOs allowed the use of H_2_O_2_ as a co-substrate for the oxidation of target compounds that produce detectable chromogenic molecules. This can be exploited for the oxidation of 2,6-DMP to coerulignone in the presence of H_2_O_2_, which can be detected at 469 nm [22,38]. The reaction was followed at concentrations of 1.25–100 mM 2,6-DMP and 0.02–20 mM H_2_O_2_, and the use of higher substrate concentrations produced strong enzyme inactivation. By using this method, the kinetic constants for both 2,6-DMP and H_2_O_2_, were determined. 2,6-DMP *K*_m_ measurements were performed by setting the H_2_O_2_ concentration at 2 mM, and for H_2_O_2_
*K*_m_ measurements, the 2,6-DMP concentration was fixed at 10 mM (Table 1). The three variants of TamAA9A have better kinetic constants, with a higher affinity (lower *K*_m_) and *k_cat_* values for both substrates than those detected for the LPMO from *N. crassa* [22], when performing the reaction at pH 6.

Regarding the three LPMO variants of *T. amestolkiae*, TamAA9A-WT and TamAA9A-CD were shown to have very similar activity and substrate specificity, while TamAA9A-CBM displayed three-fold less activity in the kinetics of 2,6-DMP, and a worse *K*_m_ in the kinetics of H_2_O_2_. This is consistent with the reduced thermostability observed for this enzyme, and is a sign that, probably, the addition of the CBM affected the activity of the enzyme and even its correct folding. Additions or eliminations of CBMs have been reported to alter the enzymatic properties of some glycosyl hydrolases [49] and LPMOs. For example, deletion of the CBMs of two AA10 LPMOs halved their activity on crystalline cellulose and PASC [50,51], and removal of the CBMs of the LPMOs from the fungi *Podospora anserina* and *N. crassa* reduced their activity and, interestingly, misbalanced their regioselectivity [52,53]. In any case, as it has been postulated before, the need to conduct a detailed analysis of the role of CBMs in LPMO action is evident [17].

In order to determine if the three LPMO variants interact with crystalline cellulose, adsorption tests were performed. Surprisingly, none of the variants bound significantly to crystalline cellulose, not even the TamAA9A-CBM chimera, which has a cellulose binding domain that is functional in a β-glucosidase from the same fungus [28]. The fact that none of these LPMOs can bind to crystalline cellulose is intriguing and should be explored in the future. Other fungal LPMOs have demonstrated their ability to bind to crystalline cellulose even lacking a CBM, as is the case for LPMO9F from *N. crassa* [15].

### 2.4. Phylogeny, and Homology Modeling of TamAA9A

To analyze the structural features of the wild-type LPMO from *T. amestolkiae*, a molecular model was built as described by below. Regarding the catalytic domain, present in the three variants, the model revealed an overall structure similar to those from other AA9LPMOs. Its fold consisted of a β-sandwich with two β-sheets, a total of eight β-strands (four per sheet) and three loops (Figure 3A). The protein contained two putative disulfide bonds between Cys56-Cys178 and Cys97-Cys101.

The first one is conserved among AA9s, connecting the L2 to the eighth β-strand, and the second one occurs in many AA9LPMOs but is not fully conserved. The histidine brace where the catalytic copper is coordinated (Figure 3B) is formed by His1 and His85 in the equatorial plane, and the axial Tyr175 residue is buried in the structure. Concerning the environment around the copper atom, TamAA9A showed a flat surface typical for AA9LPMOs able to act on crystalline substrates, without ridges or protuberant loops. LPMOs show a large variance in their substrate-binding surfaces located in the loops that connect the strands of the β-sandwich (highlighted in Figure 3C and Figure 4), defined as L2, L3 (only present in C4 oxidizers), LS (loop short) and LC (C-terminal loop) [54]. Similarly to other C1-C4 oxidizers, such as TaAA9A, the LPMO9A from *T. amestolkiae* contains the loops L2, LS and LC.

Regarding the structural elements that have been proposed to control the regioselectivity in AA9s, TamAA9A contains a Pro83 in agreement with other C1–C4 oxidizers, which would provide intermediate access to the axial position of copper among strict C1 oxidizers (with a Tyr in the homologous position) or strict C4 oxidizers (showing an Ala) [52]. However, this function is under debate and should be explored carefully, since mutagenesis studies in PcAA9D point to a more important role for the hydrophobic residues exposed in the substrate-binding surface [55].

On the other hand, the linker-like sequence from TamAA9A-WT was modeled by AlphaFold2.0 or RoseTTAFold as an extended and unstructured C-t, with poor accuracy in both cases (Appendix A). The model of the chimeric variant with a CBM, presents both domains accurately folded and bound by the linker, which is displayed either as a part of the catalytic AA9 domain by AlphaFold2.0 or as a separate domain by RoseTTAFold.

Finally, a protein phylogeny was analyzed using the sequences of characterized AA9 LPMOs available in the CAZy database. The resulting cladogram placed TamAA9A in the C1/C4 oxidizers clade (Figure 5), with the LPMOs from *T. aurantiacus* and *Aspergillus tamarii* being its closest homologues. To the best of our knowledge, the regioselectivity of fungal LPMOs from *Penicillium* or *Talaromyces* species has not been previously analyzed, but all the phylogenetic analyses usually place them as C1/C4 oxidizers [56]. To confirm the regioselectivity of TamAA9A, the reaction products were characterized by HPAEC-PAD.

### 2.5. PASC and CMC Degradation Analysis by HPAEC-PAD

For experimental confirmation of the activity and regioselectivity of the wild-type LPMO TamAA9A-WT from *T. amestolkiae*, the release of native and oxidized cello-oligosaccharides was qualitatively studied using PASC and CMC as substrates, and 1 mM ascorbic acid as the electron donor. Samples of these reactions were analyzed by HPAEC -PAD. By using this technique, reaction products could be separated according to their oxidation degree and length. In the conditions used in this work, monooxidized products elute first, followed by dioxidized products. The chromatograms of samples from enzymatic treatments of PASC and CMC are depicted in Figure 6A,B, respectively. Native cello-oligosaccharides with a degree of polymerization (DP) from two to five eluted in the first place, between 30 and 40 min. Then, small peaks that may correspond to C1-oxidized or C4-oxidized cello-oligosaccharides were detected after around 60 min, and later, at retention times of about 90 min, a final series of peaks eluted. These usually correspond to doubly oxidized products, which confirms the C1/C4 regioselectivity of TamAA9A-WT. The chromatograms showed profiles similar of those observed for other treatments with LPMOs [13].

Differences between PASC and CMC degradation can be observed mainly in the size of the peaks eluting at about 60 min (which is clearly more intense in PASC), and in the region of products with a double oxidation, that present many more peaks in CMC degradation. Moreover, the big difference found in the size of the peaks of native oligosaccharides compared with those of the oxidized species could be related to the spontaneous formation of native oligosaccharides, as explained by Westereng et al. [13]. In any case, besides the doubly oxidized products, the chromatogram showed peaks corresponding to oxidized products only in C1 or C4, which merits further investigation. It should be remarked that, as analyzing the regioselectivity of AA9 LPMOs produced by species from *Penicillium* or *Talaromyces* is a difficult task, authors usually do not report the C1 or C4 preference of the enzymes [46,56,57]. This work is expected to lay the foundations for developing further research in the AA9LPMOs produced by these genera, well known for their ability to secrete a high number of GHs involved in lignocellulose degradation, and to establish the role of these oxidative enzymes in plant biomass valorization.

### 2.6. Effect of TamAA9A in Saccharification of Cellulosic Substrates

The conversion of plant residues to glucose is a crucial step in the production of biofuels from lignocellulosic biomass. To determine if the three variants of *T. amestolkiae* LPMOs can improve the saccharification yield, four different substrates were assayed: (i) crystalline cellulose, (ii) PASC, (iii) wheat straw slurry (steam explosion pretreated), and (iv) brewers spent grain. After pretreatment of the different materials with the LPMOs (16 h), no glucose release was detected.

Then, the samples were treated with a cellulolytic commercial cocktail (Celluclast 1.5 L), following the glucose released for 40 h. In all cases, the three LPMO variants increased the final saccharification yield (Figure 7). Crystalline cellulose and PASC are cellulosic model substrates in which LPMOs should act without being inhibited by other components of the reaction mixture. The best saccharification yields were obtained with TamAA9A-WT, that increased the glucose release by around 20% for PASC and 27% for crystalline cellulose, in a significantly different manner, when compared with control samples not treated with LPMOs (Figure 7A,B). TamAA9A-CD was less efficient, increasing the yield of glucose by 13% for both substrates, and TamAA9A-CBM only enhanced this yield by 2%–3% (which was expected, since its activity was three times lower on 2,6-DMP oxidation, which may imply that the LPMO is being be inhibited by the CBM). Thus, the wild-type protein was the most effective against these substrates, and it was especially relevant in the saccharification of crystalline cellulose, overpassing by 15% the values obtained using TamAA9A-CD, while the enhancement observed for the other substrate was around 7%–9%. Although further research is needed, these data suggest that the “linker-like” region of the wild-type enzyme helps in the degradation of pure crystalline cellulose.

Finally, we evaluated if LPMOs were able to act on lignocellulose substrates, which are the materials used in real industrial processes (Figure 7C,D). Again, pretreatment with the wild-type LPMO produced the best improvements (with statistically significant differences) in the saccharification yields of these materials. The most important effect was observed for brewers spent grain, with an enhancement of the glucose released of 28% against the 17% determined for wheat straw slurry. It should be noticed that the slurry represents a complex and recalcitrant material obtained by the steam explosion of wheat straw. This severe physicochemical pretreatment is known to produce compounds that may act as inhibitors of LPMO activity. The improvements in the saccharification of these substrates when pretreated with TamAA9A-CD and TamAA9A-CBM amounted to 18% and 5% for brewers spent grain, and 8% and 4% for wheat straw slurry, respectively.

The moderate increase in the saccharification yields of all the materials assayed by pretreatment with LPMO variants of *T. amestolkiae* is in agreement with the results obtained with other fungal LPMOs [45,58,59], that enhanced saccharification of bagasse, rice straw, sorghum stover or pretreated oak by 15%–30%.

## 3. Materials and Methods

### 3.1. Microorganism and Culture Conditions

The *T. amestolkiae* A795 strain is deposited in the IJFM collection, at the Centro de Investigaciones Biológicas Margarita Salas (Madrid, Spain). The fungus was grown on PDA (potato dextrose agar) plates at 28 °C and maintained at 4 °C. For RNA extraction, the fungus was cultured in Mandels medium [28], with 1% crystalline cellulose (Merck, Darmstadt, Germany) for enzyme induction. *Escherichia coli* DH5α was selected for plasmid propagation. After transformation, it was grown at 37 °C on LB agar plates (10 g/L tryptone, 5 g/L yeast extract, 10 g/L NaCl, and 15 g/L agar) containing 25 mg/L zeocin for the selection of recombinant clones. The heterologous expression of TamAA9A was carried out using the *P. pastoris* X33 strain and the procedure was adapted from the EasySelect™ Pichia Expression Kit manual (Invitrogen, Waltham, MA, USA). The strain was cultivated on YPD plates (10 g/L yeast extract, 20 g/L peptone, 20 g/L glucose and 10 g/L agar). Transformants were selected and cultivated on YPD containing 100 mg/L zeocin and cultured 2–5 days at 28 °C. Recombinant protein was produced in YEPS medium (20 g/L peptone, 10 g/L yeast extract, 10 g/L sorbitol, and 100 mM potassium phosphate buffer, pH 6), with daily addition of 10 mL/L of methanol as an inducer. It was also performed using 5-L fermenters with a basal salt medium (phosphoric acid 26.7 mL/L, calcium sulfate 0.93 g/L, potassium sulfate 18.2 g/L, magnesium sulfate-7H_2_O 14.9 g/L, potassium hydroxide 4.13 g/L, glycerol 40.0 g/L) according to Invitrogen *Pichia* Fermentation Process Guidelines (Invitrogen, Waltham, MA, USA).

### 3.2. Identification of the T. amestolkiae LPMO Gene, Nucleic Acid Isolation, Cloning and Expression in P. pastoris

A putative LPMO nucleotide sequence, annotated as AA9A in the *T. amestolkiae* genome (GenBank accession no. MIKG00000000) [24], was identified and analyzed using the SignalP 5.0 server to identify potential signal peptides. To amplify this putative LPMO sequence (named as TamAA9A) and to detect the presence of introns, RNA was extracted with Trizol (Ambion, Waltham, MA, USA) [60] from 7-day old *T. amestolkiae* cultures growing on 1% crystalline cellulose as a carbon source. After RNA extraction, the isolated transcripts were transformed into cDNA using the Superscript II Reverse Transcriptase RT-PCR kit (Invitrogen, Waltham, MA, USA), according to the manufacturer’s instructions. Finally, PCR amplifications were performed in a thermocycler Mastercycler pro S (Eppendorf, Hamburg, Germany). Primers for TamAA9A amplification were designed by analyzing the nucleotide sequence. Restriction sites for *Xho*I and *Not*I (New England Biolabs, MA, USA) were included in the forward and reverse primers, respectively (LPMOFWXHOI: 5′-ATCTCGAGAAAAGACATGGTTATGTGCAAAACATCG-3′, and LPMORVNOTI: 5′-ATGCGGCCGCTTAAAAGACAGTGGTGGTGATGA-3′). The PCR protocol was carried out as follows: initial denaturation at 95 °C for 5 min, followed by 30 cycles of amplification: denaturation at 95 °C for 45 s, primer annealing at 55 °C for 45 s, and elongation at 72 °C for 2 min, with 72 °C for 10 min as the final step. The PCR product obtained was introduced in the pPICZα vector (Invitrogen, Waltham, MA, USA), and cloned into *E. coli* DH5α, where it was sequenced. Different plasmids were designed for recombinant production of TamAA9A. On the one hand, the protein was expressed using the α-factor signal peptide. To do so, the complete *T. amestolkiae* protein sequence (including the natural “linker-like” region) was cloned into pPICZα, after PCR amplification of the cDNA, with the primers LPMOFWXHOI and LPMORVNOTI. On the other hand, the LPMO was expressed using its native signal peptide. For this purpose, three pPICZ-derived plasmids were designed, containing either the native LPMO sequence, only the catalytic domain of the LPMO, or the native enzyme with a cellulose binding domain. This CBM belongs to BGL-2, a previously characterized β-glucosidase from the same fungus [28]. This sequence (acacagaccccgtatggacagtgtggtggacagggctggagcggtcctacagtttgttcatccggctggacttgtaaggtgacgaatcagtggtattctcaatgcctacaatag) was added after the “linker-like” region and constructed by General Biosystems (Durham, NC, USA). All the constructions were transformed into *P. pastoris* X-33 by the lithium chloride method (according to manufacturer’s instructions), after linearization with *Sac*I (New England Biolabs, MA, USA) to increase the efficiency. Transformed colonies were grown on YPD medium plates with zeocin as a selection marker.

### 3.3. Production and Purification of LPMO

Selected clones were grown overnight in 250 mL flasks with 50 mL of YPD medium at 28 °C and 250 rpm. Then, 16 mL was used to inoculate a 2 L flask containing 400 mL of YEPS medium. Cultures were incubated at 28 °C and 250 rpm for 5 days with a daily addition of 10 mL/L methanol. The enzyme secretion of positive clones was monitored by measuring the protein concentration in the culture supernatants using Bradford reagent (Bio-Rad, Hercules, CA, USA) and comparing it with non-transformed strains used as negative controls. TamAA9A production was further confirmed by SDS-PAGE. Enzyme production was performed using a BioFlo 120 (Eppendorf, Hamburg, Germany) bioreactor and the procedure was adapted from the Pichia Fermentation Process Guidelines (Invitrogen).

Enzymes were further produced in a 5-L BioFlo 120 (Eppendorf, Hamburg, Germany) bioreactor with a starting volume of 3 L basal salt medium supplemented with 4.35 mL/L of PTM1 trace salts (Invitrogen *Pichia* Fermentation Process Guidelines) and 0.1 mM CuSO_4_. After sterilization, the pH of the medium was adjusted to 5.0 with 28% ammonium hydroxide and maintained during the fermentation. The procedure was adapted from the *Pichia* Fermentation Process Guidelines of Invitrogen. After the initial batch and fed-batch phase using glycerol as the carbon source, enzyme production was induced by the addition of methanol containing 12 mL of L-1 Pichia Trace Metal solution (PTM1) and the feed rate was adjusted to maintain a 20% oxygen saturation. Samples were taken regularly and the wet biomass, protein concentration and LPMO activity based on the oxidation of 2,6-DMP, were measured (see below). Purification of the three LPMO variants was performed using an ÄKTA Purifier FPLC system (GE Healthcare Life Sciences, Chicago, IL, USA). Cells were removed by centrifugation and the supernatant was clarified by filtration. Then, 30% ammonium sulfate was added to the supernatant, and it was applied onto a 100 mL PHE-Sepharose FastFlow resin (GE Healthcare), equilibrated with 30% ammonium sulfate in 10 mM sodium phosphate buffer, pH 6.0. After binding, proteins were eluted with a linear gradient from 30% to 0% ammonium sulfate. Fractions with LPMO activity were pooled, dialyzed against 10 mM phosphate buffer pH 6, and applied onto a 5 mL QFF HiTrap cartridge (GE Healthcare) equilibrated with the same buffer. Proteins were eluted with a NaCl gradient (from 0 to 0.4 M during the 70 mL elution). Fractions with LPMO activity were pooled, concentrated, and protein homogeneity was confirmed by SDS-PAGE.

### 3.4. Protein Quantification, Enzyme Assays and Substrate Specificity

All the reagents for the reactions were purchased from Sigma Aldrich (Burlington, MA, USA). Purified LPMOs were saturated with copper by incubation with an equimolar concentration of CuSO_4_ for 1 h at room temperature. After this, the enzymes were dialyzed against 10 mM phosphate buffer, pH 6. The protein concentration of the purified enzymes was calculated by measuring the absorbance at 280 nm using a Nanodrop spectrophotometer (Thermo Fisher Scientific, Waltham, MA, USA) and by the bicinchoninic acid assay (BCA) method.

For fast detection of LPMO activity, the oxidation of 2,6-dimethoxyphenol (2,6-DMP) to coerulignone (ε_469_ = 53,200 M^−1^ cm^−1^) was followed spectrophotometrically [22,38]. The standard reaction was performed in 100 mM sodium phosphate buffer pH 6 with 10 mM 2,6-DMP, 2 mM H_2_O_2_, and 20 µg/mL of each LPMO variant. An enzymatic activity of 1 U was defined as the amount of enzyme that transforms 1 µmol substrate/min. Kinetic constants of the purified LPMOs were determined for 2,6-DMP over a range of concentrations from 100 mM to 1.25 mM (concentration of H_2_O_2_ was set at 2 mM), and for H_2_O_2_ (in a range of concentrations from 20 mM to 0.02 mM) with the concentration of 2,6-DMP fixed at 10 mM. SigmaPlot 14.0 ((Systat Software, Erkrath, Germany) was used to fit the data to the Michaelis–Menten equation and calculate the kinetic constants.

### 3.5. Physicochemical Properties. Cellulose Binding. Circular Dichroism

The LPMO variants were subjected to *N*- and *O*-deglycosylation by using Endoglycosidase H or *O*-Glycosidase (Roche, Basil, Switzerland), according to the manufacturer’s instructions. Differences in the molecular mass before and after deglycosylation were analyzed by SDS-PAGE electrophoresis. The possible glycosylation sites of the LPMOs were analyzed with the NetNGlyc 4.0 Server (https://services.healthtech.dtu.dk/service.php?NetOGlyc-4.0, accessed on 15 December 2021).

The stability at different pH values was assayed using 2,6-DMP as the substrate and Britton–Robinson buffer (100 mM) in a pH range from 2 to 10. Temperature stability was determined after incubating the enzymes for 3 h in 100 mM sodium phosphate buffer pH 6, at temperatures ranging from 40 to 95 °C, then cooling them to 4 °C for 10 min and then rewarming them to room temperature for 5 min before measuring the residual activity by the standard 2,6-DMP assay. The T_50_ value is defined as the temperature at which the enzyme loses 50% of its activity after 3 h of incubation. SigmaPlot 14.0 was used to fit the data to a sigmoidal equation. To further study the thermostability of LPMOs, they were analyzed by circular dichroism (CD) using a JASCO J-850 spectropolarimeter (Hachioji-shi, Tokyo, Japan). The conformational stability of the three LPMO variants was determined in a temperature ramp from 20 to 95 °C. The temperature was increased at a rate of 20 °C/h. Protein unfolding was monitored at 220 nm by far-UV CD [42]. The binding affinity was evaluated using crystalline cellulose as a substrate [28]. For each purified enzyme 30 ng was mixed with 500 µL of 1% crystalline cellulose (*w/v*) in 10 mM sodium phosphate buffer pH 6. The reaction was incubated at 1,200 rpm and 4 °C for 24 h. Aliquots were taken at different times (10 min, 1, 2, 3 and 24 h) and after centrifuging the samples for 1 min at 14,000× *g*, the residual LPMO activity in the supernatants was measured.

### 3.6. Phylogenetic and Protein Structure Analyses of the T. amestolkiae LPMO

A protein cladogram was built using the information on characterized AA9 LPMOs from the CAZy database and on TamAA9A. After removal of the signal peptides and CBMs, the amino acid sequences were aligned using MAFFT [61] with the L-INS-i option. The resulting multiple sequence alignment was then used as the input for PhyML [62] to obtain the cladogram using the WAG+I+G+F model, which was revealed as the best evolutionary model for the dataset using prottest3 [63] (Whelan and Goldman (WAG) evolutionary model, with amino acid frequencies (+F), proportion of invariant sites (+I) and estimated shape parameter of the gamma distribution (+G)). The classification of C1- and C4-oxidizing LPMOs was based on sequence clusters as reported by [18]. A selection of C1, C4 or C1-C4 oxidizers was used to display conserved motifs using ESPript3 [64].

Models of the *T. amestolkiae* LPMO variants were predicted by homology-based modeling using the SwissModel server [65]. To gain insight into the “linker-like region” or the chimeric construct of TamAA9A with the CBM, models of TamAA9A-WT, TamAA9A-CD and TamAA9A-CBM were additionally built using AlphaFold2.0 [66] as implemented in the SAGA supercomputer resources at NTNU, Trondheim, Norway, or RoseTTAFold [67] as implemented in the Robetta server (https://robetta.bakerlab.org/, accessed on 15 December 2021). The best model of each variant (based on per-residue estimate confidence, pLDDT for Alphafold and CA-IDDT for RoseTTAFold) was selected for analysis.

### 3.7. Analysis of Degradation Products from Cellulose by High-Performance Anion-Exchange Chromatography Coupled with Pulsed Amperometric Detection (HPAEC-PAD)

Phosphoric acid swollen cellulose (PASC), prepared as described previously [68], and carboxymethyl cellulose were used as model substrates at a concentration of 1% (*w*/*v*) for analysis of the products released by the enzymatic treatments. Both substrates were incubated for 23 h at 30 °C with the different LPMO variants in 100 mM phosphate buffer pH 6 with 1 mM ascorbic acid as an electron donor. Aliquots (200 μL) were withdrawn at different intervals and mixed with absolute ethanol to a final ethanol concentration of 70%, precipitating the residual polysaccharide. Samples were centrifuged at 10,000 rpm for 5 min and filtered through 0.45 μm nylon filters (Cosela, Seville, Spain). Then, the samples were desiccated in a vacuum concentrator 5301 (Eppendorf, Hamburg, Germany) at 30 °C until a volume of around 200 µL was reached. Negative controls without ascorbic acid and/or LPMO were similarly treated under the same conditions. The oligosaccharides released were analyzed by HPAEC-PAD in a Dionex ICS3000 system (Dionex, Thermo Fischer Scientific Inc., Waltham, MA, USA) consisting of an SP gradient pump, an electrochemical detector with a gold working electrode and Ag/AgCl reference electrode, and an autosampler (model AS-HV). All eluents were degassed by flushing with helium. A pellicular anion-exchange 4 × 250 mm Carbo-Pack PA-100 column (Dionex, Thermo Fischer Scientific Inc., Waltham, MA, USA) connected to a 4 × 50 mm CarboPac PA-100 guard column was used at 30 °C. The eluent was prepared with Milli-Q water, NaOH and sodium acetate. The mobile phase contained 10 mM NaOH from start to end, whilst two gradients were performed with sodium acetate. The first was an 80 min-gradient from 0 to 60 mM AcONa, and the second was from 60 to 160 mM in 20 min. Finally, the column was equilibrated back to the initial conditions. The chromatograms were analyzed using Chromeleon 7.2 software (Thermo Fisher, Waltham, MA, USA) and the identification of the different carbohydrates was done on the basis of commercial standards when available.

### 3.8. Saccharification of Lignocellulosic Substrates

The effect of LPMO variants on saccharification efficiency was studied using different lignocellulosic substrates at a concentration of 20 mg/mL: (i) wheat straw slurry (steam explosion pretreated), kindly provided by Abengoa (Sevilla, Spain); (ii) brewers spent grain, kindly provided by Heineken (Amsterdam, The Netherlands); (iii) crystalline cellulose (Merck, Darmstadt, Germany) and (iv) PASC. Substrates were pretreated with the LPMO variants (final concentration 1 mg/mL) for 16 h in 100 mM sodium acetate buffer pH 5 with 1 mM ascorbic acid. After incubation, each pretreated substrate was saccharified with Celluclast 1.5 L (Novozymes, Copenhagen, Denmark), a basal cocktail with low β-glucosidase activity, but rich in cellobiohydrolase and endoglucanase activities, adjusted to 1 U of β-glucosidase activity. Reactions were carried out for 24 h at 40 °C and 1200 rpm. Samples were taken at different intervals and the glucose released was quantified in the supernatants using the Glucose-TR commercial kit (Spinreact, Girona, Spain), according to the manufacturer’s instructions. Controls were treated equivalently without the previous LPMO pretreatment. All assays were performed in triplicate, and significant differences were studied using Student’s *t*-test, considering a *p* value < 0.05 as the limit for statistical differences.

## 4. Conclusions

In summary, a novel LPMO discovered in the genome of *T. amestolkiae* has been successfully expressed in *P. pastoris* and characterized. The wild-type enzyme (TamAA9A-WT), which has a “linker-like” region in the C-terminal area, was the most efficient in enhancing the saccharification yield of cellulosic and lignocellulosic substrates. The results obtained with the two enzyme variants constructed, TamAA9A-CD (only with the catalytic domain) and TamAA9A-CBM (with a fungal CBM after the “linker-like region”), suggest that the presence of the native linker favors the efficiency of the LPMO, while the incorporation of the CBM resulted in a negative effect, probably because it affects protein folding. In any case, it has been corroborated that the pretreatment of plant biomass with these enzymes helped to increase the efficiency of the process. Furthermore, it is important to note that the LPMOs tested in this work were not only able to improve the saccharification of model substrates, such as crystalline cellulose or PASC, but they were also active on natural lignocellulosic residues, such as brewers spent grain or pretreated wheat straw, which highlights the potential of these LPMOs in the framework of a circular bioeconomy context.

## Figures and Tables

**Figure 1 ijms-22-13611-f001:**
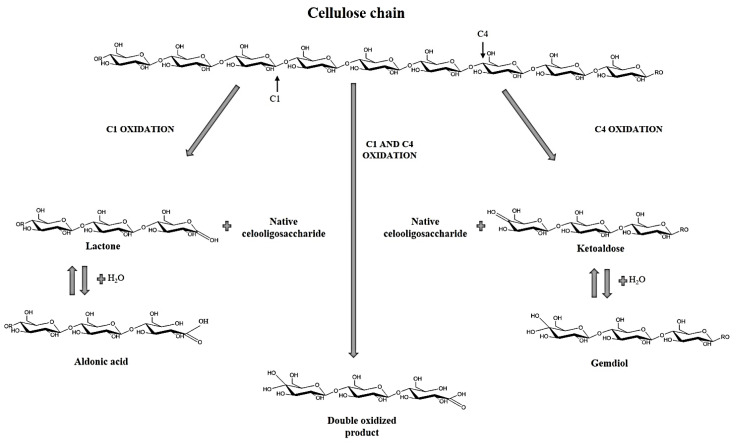
Reaction scheme of LPMO cellulose degradation, showing the products depending of the regioselectivity of the enzymes.

**Figure 2 ijms-22-13611-f002:**
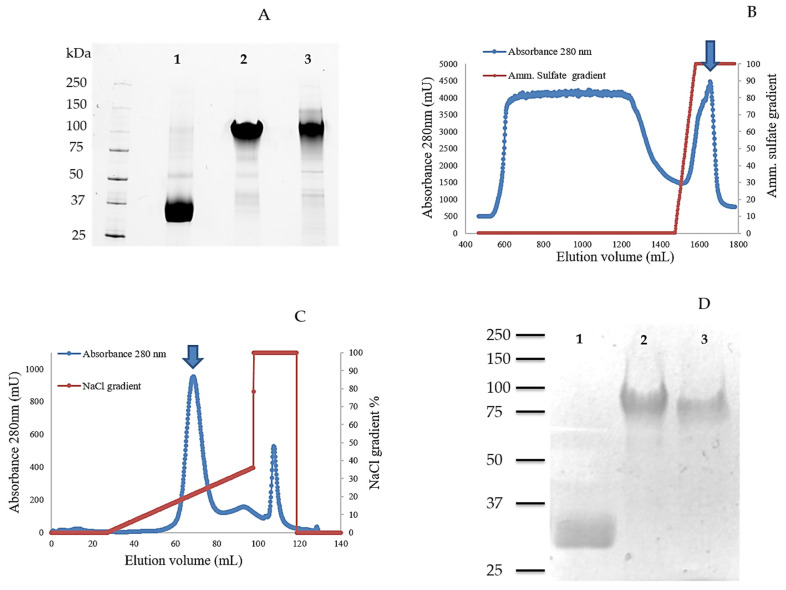
(**A**) Analysis of the expression of the three LPMO variants by SDS-PAGE. Proteins were produced in a 5-L bioreactor using Basal Salts Medium for 4 days. Lane 1: TamAA9A-CD; Lane 2: TamAA9-WT; Lane 3: TamAA9-CBM. (**B**) and (**C**) Two-step chromatographic purification of LPMOs variants from *T. amestolkiae*. The example describes the purification of TamAA9A-WT. (**B**) Hydrophobic interaction chromatography using an ammonium sulfate gradient for protein elution. (**C**) Anion-exchange chromatography, using a NaCl gradient for protein elution. Peaks containing LPMO activity are marked with arrows. (**D**) Purified LPMO variants: Lane 1, TamAA9A-CD; Lane 2, TamAA9A-CBM; Lane 3, TamAA9A-WT.

**Figure 3 ijms-22-13611-f003:**
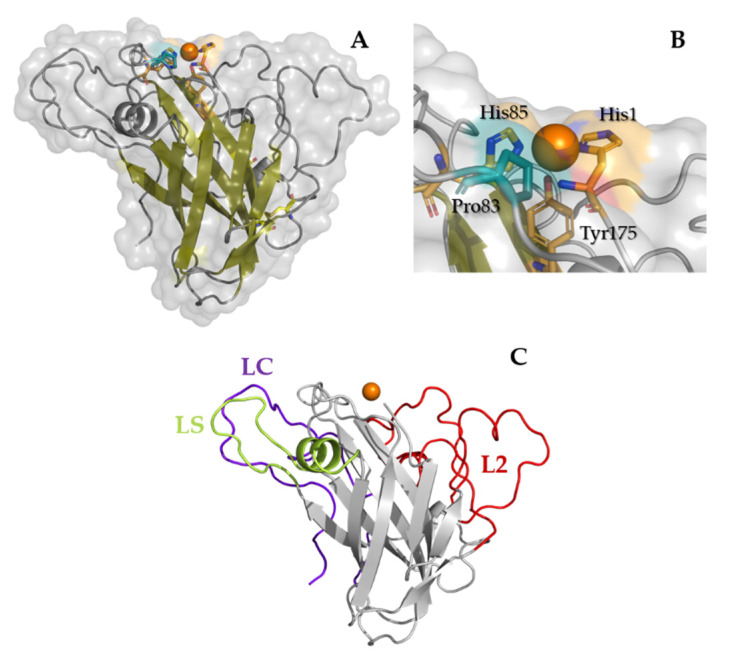
Homology model of TamAA9A built in SwissModel. The overall fold of TamAA9A is a conserved β-sandwich and displays a flat substrate-binding surface. (**A**) Residues coordinating the catalytic copper are displayed in bright orange, disulfide bonds in yellow and the putative residue controlling the accessibility to the copper and regioselectivity in deep teal. (**B**) Zoom of the catalytic center. (**C**) The loops interconnecting the strands that define the substrate-binding surface are: red, L2; green, LS; purple, LC.

**Figure 4 ijms-22-13611-f004:**
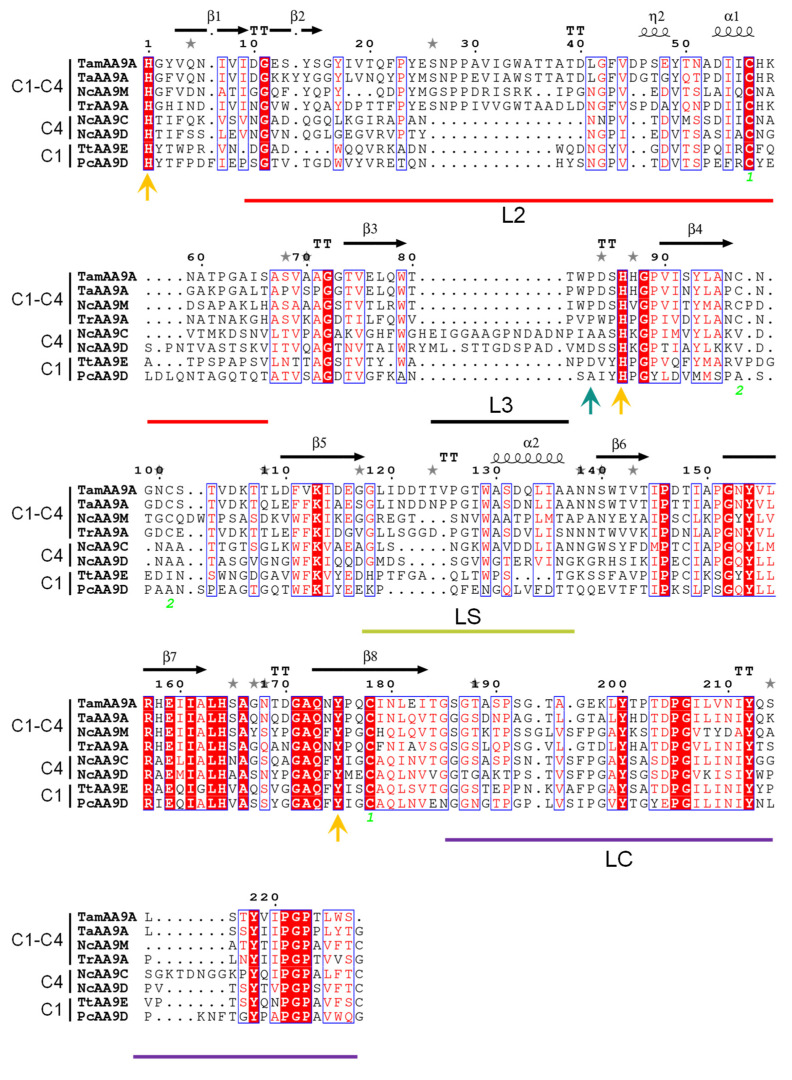
Sequence alignment of selected LPMOs. White in red background: fully conserved residues; blue frames: more than 70% of the residues aligned have similar properties (red residues on a white background). Yellow arrows indicate residues coordinating the catalytic copper. The green arrow shows the putative residue that controls regioselectivity in AA9s. Green numbers mark disulfide bonds. The loops interconnecting the strands of the β-sandwich are marked as L2, L3, LS and LC. Alignment was built with AA9 LPMOs from *Thermoascus auranticus* (TaAA9A), *Neurospora crassa* (NcAA9M, C and D), *Trichoderma reesei* (TrAA9A), *Thermothielavioides terrestris* (TtAA9E), and *Phanerochaete chrysosporium* (PcAA9D).

**Figure 5 ijms-22-13611-f005:**
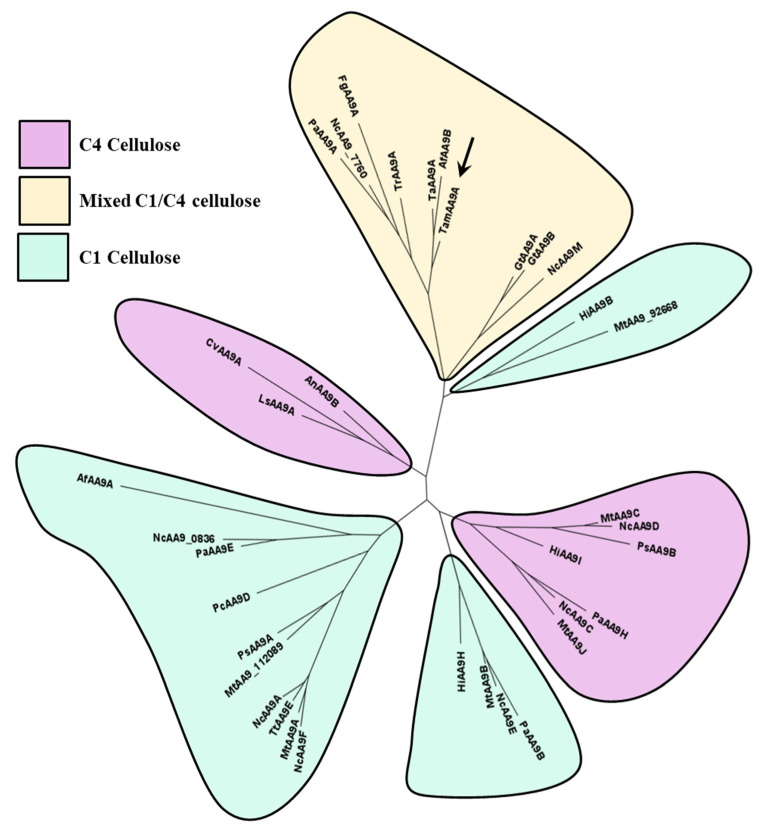
Cladogram built using the characterized AA9s in CAZy, after sequence alignment with MAFFT and tree building with PhyML. Af, *Aspergillus fumigatus*; An, *Aspergillus nidulans*; At, *Aspergillus tamarii;* Cv, *Collariella virescens*; Fg, *Fusarium graminearum*; Gt, *Gloeophyllum trabeum*; Hi, *Heterobasidion irregulare*; Ls, *Lentinus similis*; Mt, *Myceliophtora thermophila*; Nc, *Neurospora crassa*; Pa, *Podospora anserina*; Pc, *Phanerochaete chrysosporium*; Ps, *Pestalotiopsis* sp.; Ta, *Thermoascus auranticus*; Tam, *Talaromyces amestolkiae*; Tr, *Trichoderma reesei*; Tt, *Thermothielavioides terrestris*.

**Figure 6 ijms-22-13611-f006:**
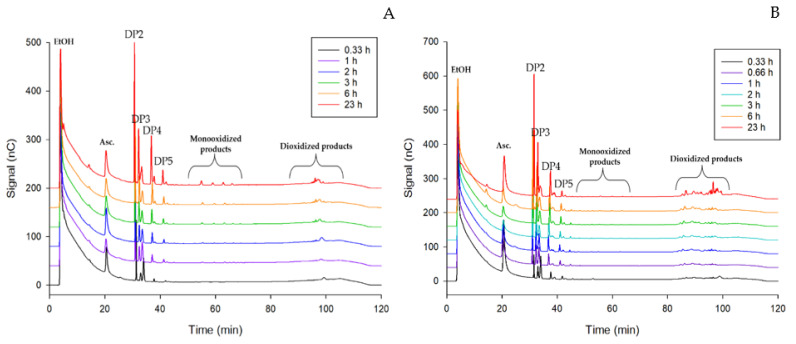
(**A**) HPAEC-PAD elution profile of PASC degradation samples after incubation with TamAA9A-WT. Reaction conditions: 1% (*w*/*v*) phosphoric acid swollen cellulose (PASC), 100 mM phosphate buffer (pH 6.0), 1 mM ascorbic acid, 20 µg/mL of LPMO, and 30 °C. (**B**) HPAEC-PAD elution profile of CMC degradation samples after incubation with TamAA9A-WT. Reaction conditions: 1% (*w*/*v*) carboxymethyl cellulose (CMC), 100 mM phosphate buffer (pH 6.0), 1 mM ascorbic acid, 20 µg/mL of LPMO, and 30 °C. Structures of the potentially obtained products can be seen in the Figure 1 in the introduction section.

**Figure 7 ijms-22-13611-f007:**
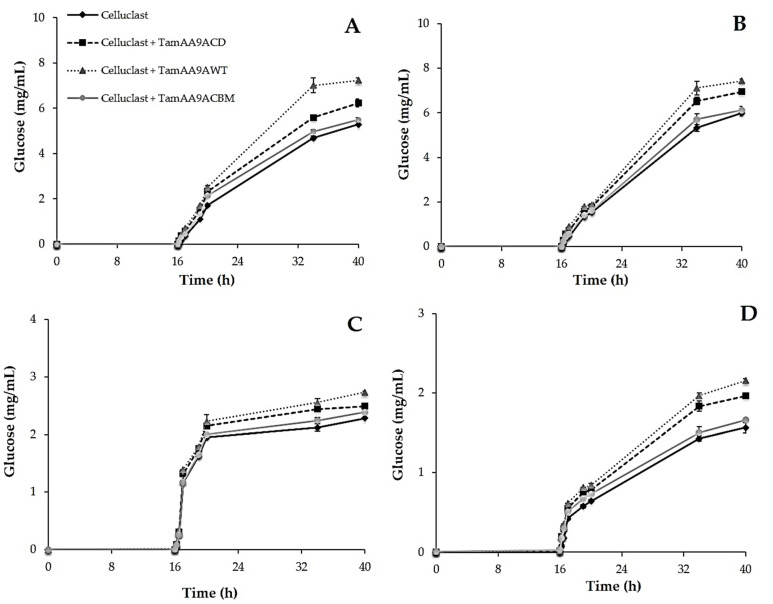
Glucose released by saccharification of different substrates pretreated for 16 h with TamAA9A-WT, TamAA9A-CD, or TamAA9A-CBM. A 20 mg amount of each material was incubated for 40 h with 1 U of β-glucosidase activity of Celluclast 1.5L, at 50 °C and 1,200 rpm. (**A**) Crystalline cellulose. (**B**) PASC. (**C**) Wheat straw slurry. (**D**) Brewers spent grain. All reactions were performed in triplicate.

**Table 1 ijms-22-13611-t001:** Kinetic constants of TamAA9A-WT, TamAA9A-CD, and TamAA9A-CBM for H_2_O_2_ and 2,6-DMP.

Enzyme	Substrate	*K_m_* (mM)	*k_cat_* (s^−1^)	*k_cat_/K_m_* (s^−1^/mM)
TamAA9A-WT	2,6-DMP	109.8 ± 0.5	0.68 ± 0.03	0.006
	H_2_O_2_	0.28 ± 0.01	0.14 ± 0.01	0.501
TamAA9A-CD	2,6-DMP	87.7 ± 0.7	0.96 ± 0.04	0.011
	H_2_O_2_	0.31 ± 0.02	0.17 ± 0.01	0.548
TamAA9A-CBM	2,6-DMP	85.68 ± 0.7	0.53 ± 0.02	0.003
	H_2_O_2_	1.03 ± 0.06	0.25 ± 0.03	0.135

## Data Availability

T. amestolkiae whole genome shotgun sequencing project is available at https://www.ncbi.nlm.nih.gov/nuccore/MIKG00000000 (accessed on 15 December 2021).

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
