# Peer review of "Lytic Polysaccharide Monooxygenase from *Talaromyces amestolkiae* with an Enigmatic Linker-like Region: The Role of This Enzyme on Cellulose Saccharification"

_ijms, 2021, doi:10.3390/ijms222413611_

Round 1
Reviewer 1 Report
Dear authors article entitled “Lytic polysaccharide monooxygenase from Talaromyces amestolkiae with an enigmatic linker-like region: the role of this enzyme on cellulose saccharification”, is interesting with significant novelty impact. The manuscript is well written, however, some details need improvement before eventual publication in IJMS.
Comments;
- Why authors not used a HIS6 tag to simplification of protein purification process? The pPICZα vector allows for the introduction of this tag on the C-terminal end of recombinant protein.
- Figure `1. The axis Y, in figure 1b should be corrected to ammonium sulfate gradient. Moreover, the washing stage of the elution profile could be shown with an axis brake.
- The catalytic efficiency expressed as the ratio of Kcat to Km should be added to presented in table 1 result of kinetic constants.
- The straight lines at SDS-PAGE presented in figure 1D suggest that the analysis was not performed at one time, as it is presented in figure 1A. This kind of manipulation is not allowed.
- Figure 4. Some labels overlap. Please correct.
Author Response
We appreciate the perception of reviewer 1 about our paper, and its comments to further improve the manuscript.
Comments
1. Why authors not used a HIS6 tag to simplification of protein purification process? The pPICZα vector allows for the introduction of this tag on the C-terminal end of recombinant protein.
Although the addition of a His tag to a protein strongly simplify the purification procedure, we have not been successful when applying this procedure to P. pastoris recombinant proteins. This could be due to the hyperglycosylation of heterologous expressed enzymes in this yeast. Also, histidines are of utmost importance for correct copper coordination of LPMOs active center, and thus, His tag could interfere with protein proper folding.
2. Figure `1. The axis Y, in figure 1b should be corrected to ammonium sulfate gradient. Moreover, the washing stage of the elution profile could be shown with an axis brake.
We apologize for this mistake; we have prepared a new version of the graphic.
3. The catalytic efficiency expressed as the ratio of Kcat to Km should be added to presented in table 1 result of kinetic constants.
We agree with the reviewer. We have replaced Vmax by kcat in Table 1.
4. The straight lines at SDS-PAGE presented in figure 1D suggest that the analysis was not performed at one time, as it is presented in figure 1A. This kind of manipulation is not allowed.
First of all, we apologize for the confusion. We rearranged the gel lanes to follow the same order than in figure 1A, but we have changed Figure 1D including the original gel picture.
5. Figure 4. Some labels overlap. Please correct.
Figure 4 has been improved to avoid the overlap of the labels.
Reviewer 2 Report
The manuscript describes the expression and characterization of a novel LPMO of the collulolytic fungus Talaromyces amestolkiae. The enzyme and its variants were expressed in Pichia pastoris and extensively characterized. Enzymatic hydrolysis of lignocellulosic biomass is of utmost importance in current biorefineries and the discovery of new enzymes are always a valuable contribution to the field. However, the authors should better highlight what are the main features of this work that make this new LPMO different or more interesting from other previously known enzymes of this family.
MINOR POINTS
- Some symbols (many beta) are missing in the text
- What are the main features of the AA9 family?
- In the introduction, a reaction scheme of the LPMO stressing the different regioselectivity and co-substrates would be helpful
- More details should be provided for the methods described at paragraph 2.6. For example, what is the meaning of WAG+I+G+F?
- Lines 292-293. Which beta-glucosidase was used?
- Lines 337-339. The sentence is not clear. What are the additional aminoacids?
- Table 1 is repeated twice
- Kinetic characterization. The kinetic parameters of the first substrate were not determined at saturation levels of the other substrate. Moreover, also the concentration range of 2,6-DMP (1.25-100 mM) does not appear appropriate being some of the determined Km near to its upper limit (and sometimes beyond it)
- Do the author can make a hypothesis about the Km of the actual substrates of the enzymes?
- Some of the structural features of the enzyme, suggested by the bioinformatic analysis, for example the role of Pro83, could have been tested experimentally
- Paragraph 3.5. The presented technique and analysis is in my opinion very hard to understand from a non-expert of the field. The authors should try to explain the basis of the technique and of the results they have. Also, the introduction in Figure 5 of structures illustrating monoxidized and dioxidized products would be helpful.
- Figure 6 and corresponding text. It is not clear from the figure if the increase in the saccharification exerted by the LPMO is statistically significant.
Author Response
We really thank the positive thoughts of reviewer about our work, and all the comments for improving the final quality of the paper. Regarding the special features of this enzyme, its kinetic values against DMP are much better than those described for others, as included in the text (lines 452-453).
MINOR POINTS
- Some symbols (many beta) are missing in the text
The manuscript has been carefully checked in order to detect misspellings.
1. What are the main features of the AA9 family?
LPMOs are distributed in seven families (AA9, AA10, AA11, AA13, AA14, AA15 and AA16) in the auxiliary activities classes of CAZy based on sequence similarity. But, despite this variety, only a few LPMOs have been characterized until date, mostly from families AA9 and AA10. While AA10 family is more complex in terms of origin and substrate specificity (members of AA10 are derived from archaea, bacteria, fungi or viruses) and have activity on different substrates like chitin and cellulose, the enzymes of AA9 are cellulose-active LPMOs found exclusively in fungi. Recently, some AA9 LPMOs active on xyloglucan has been discovered, but, in any case, they seem to be specific of β-1,4 linked glucoses. Also AA9 proteins are usually glycosylated enzymes and, the histidine of the N terminal part of the enzyme is methylated. AA9 and AA10 proteins physicochemical properties are similar, showing high thermostability.
In any case, much research is needed to further discover the different characteristics that define each family of LPMOs. A sentence has been added in the introduction section (pag 2, line 57-59).
2. In the introduction, a reaction scheme of the LPMO stressing the different regioselectivity and co-substrates would be helpful
We appreciate reviewer 2 suggestion, we have added the required figure (Fig 1), and also some phrases in the introduction, expanding the explanation of the possible degradation products of these enzymes (lanes 79-85).
3. More details should be provided for the methods described at paragraph 2.6. For example, what is the meaning of WAG+I+G+F?
We have added a paragraph that expand the explanation of this section. Lane 258-260.
4. Lines 292-293. Which beta-glucosidase was used?
We have used Celluclast 1.5L, a commercial cocktail containing several cellulases enzymes from Trichoderma reesei, including β-glucosidase from this fungus, at low but detectable level. We set BGL activity added to 1 U, to ensure that it was enough for the saccharification process.
5. Lines 337-339. The sentence is not clear. What are the additional aminoacids?
We apologize for the unclear sentence. In some cases, a very high over expression of enzymes in P. pastoris leads to an incorrect processing of N-terminal sequence, resulting in signal peptide misscleavage. Thus, some proteins retain aminoacids from the signal peptide sequence in the N terminal part. This was studied in our laboratory by Vaquero et al., (2015) in a lipase/sterol esterase heterologous expression. This is just a hypothesis to explain why LPMOs were not successfully produced after α-factor, and need to be explored in the future. Sentence has been rewritten in order to improve clarity (lanes 353 to 358).
Vaquero ME, Barriuso J, Medrano FJ, Prieto A, Martínez MJ. Heterologous expression of a fungal sterol esterase/lipase in different hosts: Effect on solubility, glycosylation and production. J Biosci Bioeng. 2015 Dec;120(6):637-43.
6. Table 1 is repeated twice
This mistake has been corrected.
7. Kinetic characterization. The kinetic parameters of the first substrate were not determined at saturation levels of the other substrate. Moreover, also the concentration ranges of 2,6-DMP (1.25-100 mM) does not appear appropriate being some of the determined Km near to its upper limit (and sometimes beyond it)
We appreciate reviewer suggestion, and we are aware about this problem.
We had issues when combining higher concentration of substrates, that lead to enzyme inactivation. When 2,6-DMP concentrations above 100 mM were used, completely inactivation of enzyme kinetics occurred in seconds. Also, 2,6-DMP is quite insoluble.
In any case, the kinetic obtained showed a Michaelis Menten behavior.
8. Do the author can make a hypothesis about the Km of the actual substrates of the enzymes?
Since the real substrate of the enzyme is a polysaccharide, it is very complicated to calculate kinetic constants. Beside the nature of the substrate, usually products are under detected. It could be expected that Km of actual substrates of the enzyme was quite lower than that obtained with synthetic ones, but this is a question unanswered by the time in the literature.
9. Some of the structural features of the enzyme, suggested by the bioinformatic analysis, for example the role of Pro83, could have been tested experimentally
We agree that mutagenesis studies on Pro83 would be very useful for enzyme characterization. We will consider it for future works.
10. Paragraph 3.5. The presented technique and analysis is in my opinion very hard to understand from a non-expert of the field. The authors should try to explain the basis of the technique and of the results they have. Also, the introduction in Figure 5 of structures illustrating monoxidized and dioxidized products would be helpful.
HPAEC-PAD is a technique that allows product separation according to the oxidation state and the length of the oligosaccharides. In the conditions used in this work, monooxidized products elute first, followed by dioxidized products.
A new sentence has been added (lanes 541 to 543), and references to figure 1 (where products formation is explained) are inserted in the figure 6 legend.
11. Figure 6 and corresponding text. It is not clear from the figure if the increase in the saccharification exerted by the LPMO is statistically significant.
Statistically significant differences are found, using T-student, between the TamAA9-WT pretreated samples and all the others. We also find differences between TamAA9-CD preatreated samples and TamAA9-CBM pretreated and non pretreated samples. Unfortunately, no differences were detected between TamAA9-CBM pretreated and non pretreated samples. We have added some phrases to clarify this in 2.8 and 3.6 section (lines 311-313 and 604-605, respectively).